# Human pannexin 1 channel is not phosphorylated by Src tyrosine kinase at Tyr199 and Tyr309

**Zheng Ruan†, Junuk Lee, Yangyang Li, Juan Du*, Wei Lü***

Department of Structural Biology, Van Andel Institute, Grand Rapids, United States

**\*For correspondence:**
juan.du@vai.org (JD);
wei.lu@vai.org (WL)

**Present address:** †Department of Biochemistry & Molecular Biology, Thomas Jefferson University, Philadelphia, United States

**Competing interest:** The authors declare that no competing interests exist.

**Abstract** Protein phosphorylation is one of the major molecular mechanisms regulating protein activity and function throughout the cell. Pannexin 1 (PANX1) is a large-pore channel permeable to ATP and other cellular metabolites. Its tyrosine phosphorylation and subsequent activation have been found to play critical roles in diverse cellular conditions, including neuronal cell death, acute inflammation, and smooth muscle contraction. Specifically, the non-receptor kinase Src has been reported to phosphorylate Tyr198 and Tyr308 of mouse PANX1 (equivalent to Tyr199 and Tyr309 of human PANX1), resulting in channel opening and ATP release. Although the Src-dependent PANX1 activation mechanism has been widely discussed in the literature, independent validation of the tyrosine phosphorylation of PANX1 has been lacking. Here, we show that commercially available antibodies against the two phosphorylation sites mentioned above—which were used to identify endogenous PANX1 phosphorylation at these two sites—are nonspecific and should not be used to interpret results related to PANX1 phosphorylation. We further provide evidence that neither tyrosine residue is a major phosphorylation site for Src kinase in heterologous expression systems. We call on the field to re-examine the existing paradigm of tyrosine phosphorylation-dependent activation of the PANX1 channel.

## eLife assessment

The current manuscript re-examines an established claim in the literature that human PANX-1 is regulated by Src kinase phosphorylation at two tyrosine residues, Y199 and Y309. This issue is **important** for our understanding of Pannexin channel regulation. The authors present an extensive series of experiments that fail to detect PANX-1 phosphorylation at these sites. Although the authors' approach is more rigorous than the previous studies, this work relies primarily on negative results that are not unambiguously definitive; the work nonetheless provides a **compelling** reason for the field to reexamine conclusions drawn in earlier studies.

## Introduction

PANX1 is a large-pore channel widely associated with many physiological and pathological processes, including immune response, inflammation, and vasoconstriction (*Adamson and Leitinger, 2014*; *Makarenkova and Shestopalov, 2014*; *Good et al., 2015*). The activation of PANX1 allows the release of ATP and other cellular metabolites to the extracellular space, serving as signaling molecules for neighboring cells (*Chekeni et al., 2010*; *Medina et al., 2020*). While the functional relevance of PANX1 in human physiology and diseases has been investigated for nearly two decades since the PANX family was cloned (*Baranova et al., 2004*; *Sandilos and Bayliss, 2012*; *Scemes and Velíšková, 2019*), the detailed structural and biophysical properties of the channel were only beginning to be elucidated. In 2020, we and others determined structures of PANX1 by single-particle cryo-electron microscopy

(cryo-EM), revealing the channel stoichiometry, domain architecture, ion permeation pathways, and inhibition mechanisms (*Michalski et al., 2020*; *Deng et al., 2020*; *Qu et al., 2020*; *Jin et al., 2020*; *Mou et al., 2020*; *Ruan et al., 2020*). These structures have in turn motivated small-molecule docking analysis and the development of the first PANX1-selective channel blockers (*Crocetti et al., 2021*). Despite the remarkable progress made in the PANX1 field, the dynamic regulation of the channel opening and closing in various cellular contexts is not well understood. Which cellular stimuli PANX1 responds to and how they achieve channel activation, especially in a reversible manner, are unclear and largely under debate (*Chiu et al., 2018*; *Mim et al., 2021*). Deciphering the regulatory principle of the PANX1 is not only important to better characterize the function of the channel, but may also provide clues to the development of pharmaceutical agents to treat PANX1-associated diseases.

Post-translational modifications, such as protease cleavage, glycosylation, phosphorylation, and S-nitrosylation, have been shown to play important roles in PANX1 regulation (*Boyce et al., 2018*). For example, caspase-dependent cleavage of the regulatory C-terminal tail of PANX1 is critically implicated for apoptotic cell clearance by the immune system (*Chekeni et al., 2010*; *Medina et al., 2020*). The glycosylation at the second extracellular loop of PANX1 sterically prevents the docking of two PANX1 heptamers into a gap-junction-like assembly (*Sosinsky et al., 2011*; *Dahl and Locovei, 2006*; *Huang et al., 2007*). Our recent cryo-EM study on PANX1 directly visualized both mechanisms and corroborates with prior studies (*Ruan et al., 2020*). While the caspase-dependent cleavage irreversibly activates PANX1, studies from several groups have converged on the concept that protein phosphorylation may be a novel mechanism to reversibly regulate PANX1 channel activity. In particular, the non-receptor tyrosine kinase Src seems to be a critical component during this process. For example, prolonged activation of P2X7 receptor leads to activation of PANX1 in many cell lines, opening of a large pore (*Locovei et al., 2007*; *Pelegrin and Surprenant, 2006*). Through the use of a mimetic peptide and kinase inhibitors, Src was later found to be involved in mediating the initial steps of this signal transduction, but it does not directly phosphorylate PANX1 per se (*Iglesias et al., 2008*). In a more recent study, PANX1 was proposed to make functional crosstalk with α7 nicotinic acetylcholine receptor (α7 nAChR) in neuroblastoma cell line (*Maldifassi et al., 2021*). Src kinase is also involved in this process; however, it is unclear if PANX1 is a direct substrate of Src (*Maldifassi et al., 2021*). In addition to tyrosine phosphorylation, murine PANX1 has been reported to undergo serine phosphorylation at Ser205 and Ser206, although two studies have shown contradictory effects of the two serine phosphorylation sites on PANX1 channel activity (*Poornima et al., 2015*; *Medina et al., 2021*). Likewise, the S394 of rat PANX1 could be phosphorylated by CaMKII upon an increase in cytoplasmic $Ca^{2+}$ concentration (*López et al., 2021*). However, this residue is not conserved in human PANX1 (hPANX1). More recently, T302 and S328 were reported to be phosphorylated by protein kinase A, which affects the mechanically induced chloride conductance of PANX1 through the lateral tunnels (*López et al., 2020*).

Direct evidence of PANX1 phosphorylation by Src was provided by two groups using western blot analysis, with Tyr198 and Tyr308 of mouse/rat PANX1 (equivalent to Tyr199 and Tyr309 of hPANX1) being the target residues. Interestingly, the underlying signaling events of PANX1 phosphorylation are non-canonical. For example, the phosphorylation of Tyr308 by Src kinase depends on ligand binding to the metabotropic NMDA receptor (NMDAR) in CA1 pyramidal neurons, but does not require its pore opening and $Ca^{2+}$ conductance (*Weilinger et al., 2012*; *Weilinger et al., 2016*). As a result, it was suggested that NMDARs, Src kinase, and PANX1 could form a signaling complex, in which the NMDAR is the ligand-sensing module whereas the PANX1 is the pore-forming unit. Phosphorylation at Tyr308 is a key step for PANX1 activation in the NMDAR/Src/PANX1 signalosomes that eventually leads to the neuronal cell death during ischemia or stroke. Likewise, the phosphorylation of Tyr198 of mouse PANX1 (mPANX1) was found to be relevant in several different cellular contexts. For example, *Lohman et al., 2015* showed that in vascular endothelial cells, Tyr198 phosphorylation—which depends on type-1 TNF receptor (TNFR1) and Src kinase—opens PANX1 channel, leading to ATP release and promoting leukocyte adhesion and emigration. Moreover, Tyr198 phosphorylation is linked to vasoconstriction and blood pressure regulation. Specifically, *DeLalio et al., 2019* found that Tyr198 is constitutively phosphorylated by Src in vascular smooth muscle cells, possibly through an α1-adrenergic receptor (α1-AR)-based vasoconstriction mechanism. In a more recent study, Nouri-Nejad et al. identified Tyr150 of PANX1 as another phosphorylation site, presumably by Src, through electrospray ionization mass spectrometry, but only to a small degree (in 2 out of 11 LC-MS/MS

spectra) (*Nouri-Nejad et al., 2021*). The functional consequence of Tyr150 phosphorylation to PANX1 is unclear, as phosphor-ablating mutant Y150F does not alter the properties of PANX1 current (*Nouri-Nejad et al., 2021*).

The initial aim of our study was to investigate how tyrosine phosphorylation regulates the channel activity of PANX1 through structural approaches. To this end, we first examined whether we could detect PANX1 phosphorylation using two commercially available antibodies claimed to be specific for phosphorylated PANX1 at the Y198 and Y308 sites, respectively. Surprisingly, neither antibody recognized PANX1 co-expressed with a constitutively active Src variant. This result raised concerns about the validity of the conclusion that Tyr198 and Tyr308 of PANX1 are phosphorylated, as identification of both sites relied on these antibodies. We further provided evidence that PANX1 is not phosphorylated at Tyr198 and Tyr308 by Src kinase, at least in heterologous expression systems. Therefore, the paradigm of tyrosine phosphorylation-dependent activation of PANX1 channel by Src kinase needs to be reconsidered.

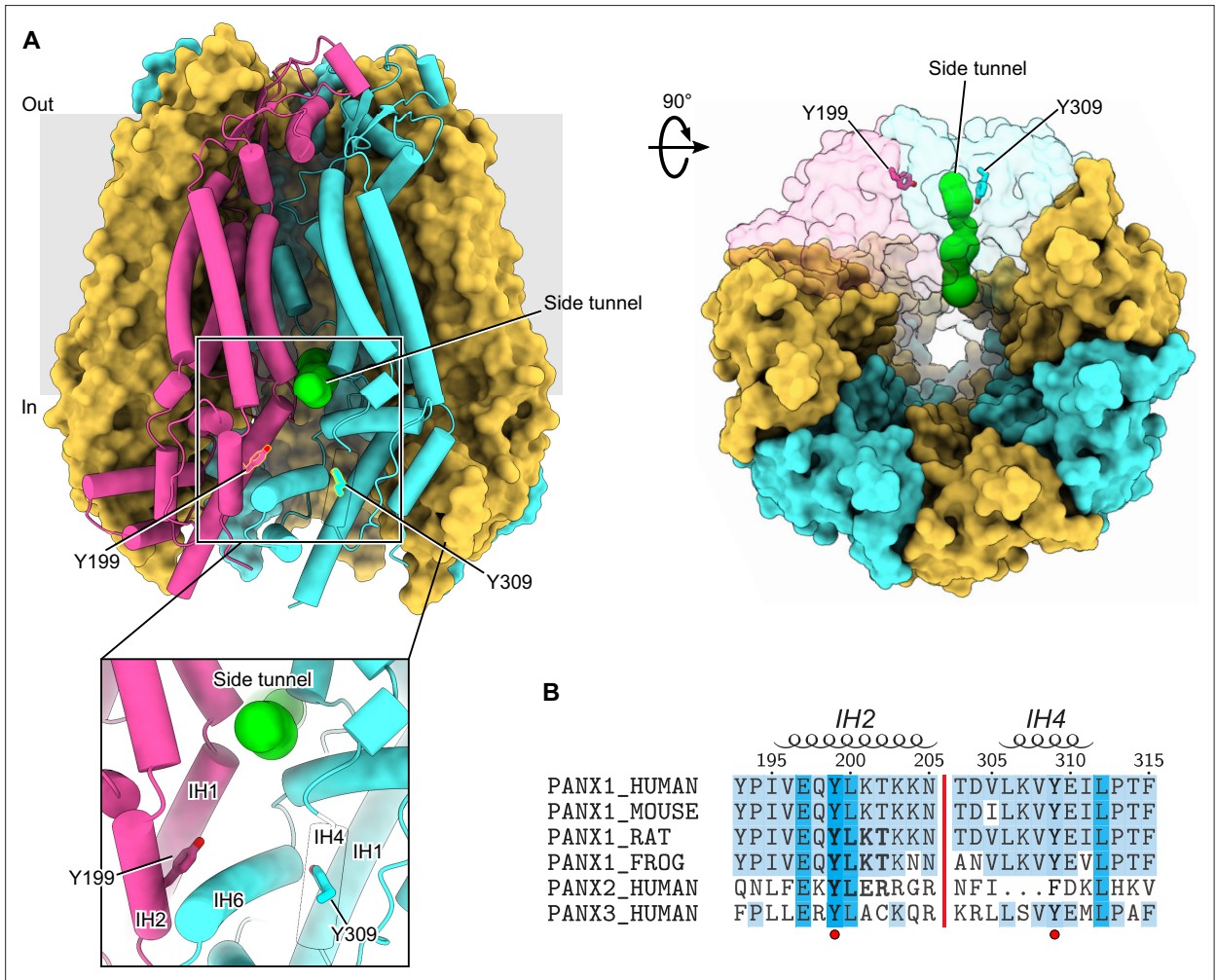

**Figure 1.** Location and sequence conservation of Tyr199 and Tyr309 in human Pannexin 1 (PANX1). (**A**) The Tyr199 and Tyr309 of human PANX1 (hPANX1) are located close to the side tunnel. Two adjacent subunits are shown in semi-transparent surface representation. The Tyr199 and Tyr309 are shown in stick representation. The green spheres indicate the pathway of the side tunnel. (**B**) Sequence alignment of PANX1 orthologs, human PANX2, and PANX3. Tyr199 and Tyr309 are highlighted using a red circle. Secondary structural features are shown on the top.

## Results

### Structural location and conservation of putative tyrosine phosphorylation sites in PANX1

Studies reporting the Src-mediated phosphorylation of Tyr198 and Tyr308 in murine PANX1 occurred prior to the availability of PANX1 structures (*Michalski et al., 2020*; *Deng et al., 2020*; *Qu et al., 2020*; *Jin et al., 2020*; *Mou et al., 2020*; *Ruan et al., 2020*; *Weilinger et al., 2016*; *DeLalio et al., 2019*). We first mapped the corresponding residues, Tyr199 and Tyr309, into the apo state structure of hPANX1 (*Figure 1A*). Interestingly, both residues are located in positions close to the side tunnel of PANX1, an anion-selective permeation pathway responsible for the PANX1 basal current (*Ruan et al., 2020*). Tyr199 is part of the intersubunit interface of PANX1 and is partially buried, which may make it less accessible to Src kinase. Tyr309, on the other hand, is completely buried inside the protein. Therefore, phosphorylation of Tyr199 and Tyr309, if occurs, would require at least conformational changes or partial unfolding of the intracellular domain. Such structural alternations are unlikely to occur spontaneously without specific cellular conditions that favor this process. Accordingly, phosphorylation of either of these two residues is expected to cause major conformational changes of the intracellular domain due to the negative charges and the bulky size of the introduced phosphate group(s).

We next compared the sequences of PANX1 across different species (*Figure 1B*). Both Tyr199 and Tyr309, as well as several residues nearby, are conserved among different PANX1 orthologs (*Figure 1B*), suggesting that phosphotyrosine antibodies developed against one species are likely to cross-react with the others. Tyr199 is also conserved in PANX2 and PANX3, the other two members of the pannexin family. In contrast, Tyr309 is only conserved in PANX1 and PANX3, while PANX2 has a phenylalanine at this position and lacks a few residues preceding this residue (*Figure 1B*). Given the critical location of the two tyrosine residues, we decided to experimentally characterize their phosphorylation status in the presence of Src kinase.

### Commercial antibodies, anti-PANX1-pY198 and anti-PANX1-pY308, do not recognize phosphorylated human PANX1 in HEK293T cells

To investigate the phosphorylation of hPANX1 by Src, we used two commercially available phosphor-specific antibodies that were used in previous studies to identify Tyr198 and Tyr308 as phosphorylation sites of murine PANX1 (*Weilinger et al., 2016*; *Lohman et al., 2015*; *DeLalio et al., 2019*) and were later reported to react with hPANX1 (*Metz and Elvers, 2022*). In addition to wild-type hPANX1, we tested phosphor-ablating mutants Y199F and Y309F (equivalent to mPANX1 Y198F and Y308F) as negative controls, which are expected to abrogate the antibody binding even in the presence of active Src kinase. As in the prior studies (*Weilinger et al., 2016*; *DeLalio et al., 2019*), we used mouse Src (mSrc), which has 99.4% pairwise sequence similarity to human Src. To increase the chance of detecting phosphorylated PANX1 and to introduce a negative control for mSrc, we generated two mSrc mutants, Y529F and K297M, which enhances and abolishes the kinase activity, respectively (*Figure 2A*). For the ease of fluorescence imaging, our hPANX1 and mSrc genes are tagged with GFP and mCherry tags, respectively (See Materials and Methods for detailed construct information), which allowed us to compare the in-gel fluorescence with the western blot signals.

From the non-transfected HEK293T cells, neither anti-PANX1-pY198 nor anti-PANX1-pY308 produced any signal, suggesting that the antibodies do not recognize endogenous proteins when hPANX1 and mSrc are absent (*Figure 2B*, lane 1). Similarly, from cells transfected with hPANX1 alone, including WT, Y199F, and Y309F, neither antibody produced any signal, despite the presence of clear GFP signals in the fluorescent gel indicating PANX1 expression (*Figure 2B*, lane 4–6). This is consistent with the notion that both antibodies do not recognize non-phosphorylated PANX1.

Unexpectedly, from another control sample of cells transfected with WT mSrc alone, anti-PANX1-pY198 yielded two prominent bands close to 75 kDa; anti-PANX1-pY308 also produced a band corresponding to the lower band of anti-PANX1-pY198, albeit much weaker (*Figure 2B*, lane 2). While the lower band corresponded to the fluorescent signal of mSrc (*Figure 2B*, lane 2), the upper band may represent endogenous proteins induced and/or phosphorylated by mSrc. The same signal pattern was detected with anti-PANX1-pY198 antibody from cells expressing the constitutively active mSrc-Y529F variant alone, although both bands were slightly upshifted (*Figure 2B*, lane 3). This upward shift was also observed in the fluorescent signal of mSrc-Y529F, suggesting that it may

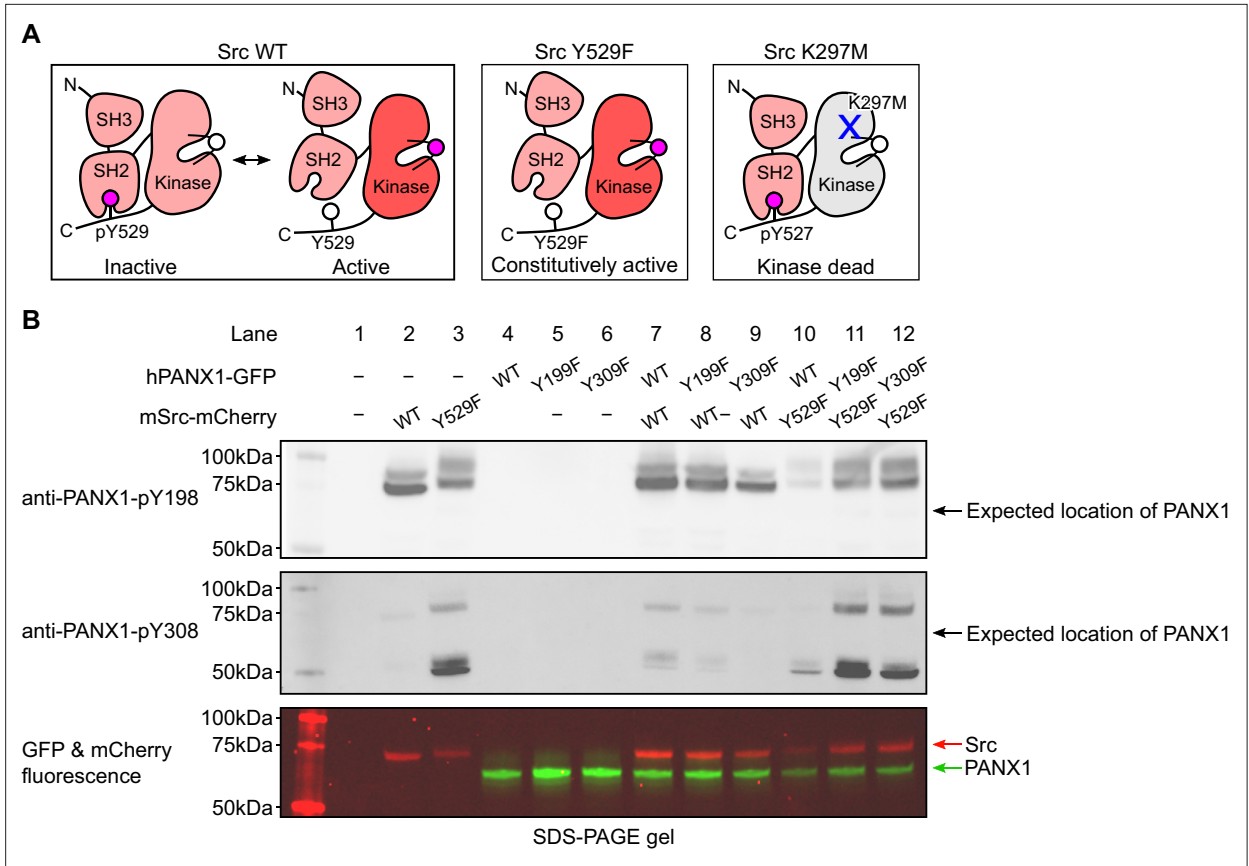

**Figure 2.** Commercially available phospho-Pannexin 1 (PANX1) antibodies (anti-PANX1-pY198 and anti-PANX1-pY308) do not recognize PANX1. (**A**) A schematic figure showing different mSrc mutants used in the study. Mouse Src wild-type (WT) undergoes a dynamic equilibrium between the inactive and active states that depends on the phosphorylation status of Tyr529. The Y529F mutation renders Src constitutively active due to the disruption of the autoinhibitory interaction between the SH2 domain and the C-terminal tail (**Nada et al., 1991**). The K297M is catalytically incompetent as the mutation abolishes the ATP-binding capability of mSrc (**Roskoski, 2015**). (**B**) Human PANX1 (WT, Y199F, or Y309F) are co-expressed with mSrc (WT or Y529F) in HEK293T cells. The cell lysates are analyzed by SDS-PAGE gel and blotted with anti-PANX1-pY198 (top) and anti-PANX1-pY308 (middle) antibodies. The in-gel fluorescence of GFP and mCherry signal are shown at the bottom. The positions of the signals detected by anti-PANX1-pY198 and anti-PANX1-pY308 antibodies do not match the position of the GFP fluorescence signal, indicating that the two antibodies are not specific for PANX1.

The online version of this article includes the following source data and figure supplement(s) for figure 2:

**Source data 1.** Uncropped immunoblots for *Figure 2B*.

**Source data 2.** Raw image for anti-PANX1-pY198 immunoblot in *Figure 2B*.

**Source data 3.** Raw image for anti-PANX1-pY308 immunoblot in *Figure 2B*.

**Source data 4.** Raw image for in-gel fluorescence (GFP and mCherry) in *Figure 2B*.

**Figure supplement 1.** The human PANX1 (hPANX1) cannot be detected by anti-PANX1-pY198 and anti-PANX1-pY308 irrespective of C-terminal GFP tag.

**Figure supplement 1—source data 1.** The raw images for the anti-PANX1-pY198 and anti-PANX1-pY308 blots in *Figure 2—figure supplement 1*.

**Figure supplement 1—source data 2.** The raw images for the anti-PANX1-PANX1 and anti-PANX1-Src blots in *Figure 2—figure supplement 1*.

**Figure supplement 1—source data 3.** The raw image for the in-gel fluorecense in *Figure 2—figure supplement 1*.

be due to a more extensive phosphorylation of mSrc-Y529F itself (lower band) and other endogenous proteins (upper band) associated with mSrc-Y529F (*Figure 2B*, lane 3). Interestingly, in addition to the band corresponding to the mSrc fluorescence signal, the anti-PANX1-pY308 antibody showed a major band at about 50 kDa (*Figure 2B*, lane 3), which may represent other sets of endogenous proteins induced and/or phosphorylated by mSrc. Taken together, our control experiments showed that both antibodies demonstrate promiscuous activity against mSrc and/or other unknown

endogenous proteins, raising concerns as to whether these antibodies should be used to interpret the phosphorylation of PANX1.

Despite the caveats of the antibodies mentioned above, we proceeded to test whether the two antibodies could at least produce additional signals that may represent phosphorylated hPANX1 (when hPANX1 and mSrc are co-expressed). We designed several co-transfection experiments combining WT mSrc or its constitutively active variant (Y529F) with WT PANX1 or its phosphor-ablating mutants (Y199F and Y309F). All combinations yielded the same signal pattern as when mSrc was expressed alone (*Figure 2B*, lanes 7–9 vs 2, lanes 10–12 vs 3), although under some conditions the signals were weaker, which we attribute to the lower protein expression level indicated by the fluorescent signals (*Figure 2B*, lane 10). Even when the constitutively active mSrc was present, both antibodies failed to detect any signals corresponding to the GFP fluorescence signal of hPANX1 (*Figure 2B*, lanes 7–12). Furthermore, there was no difference in the signal pattern of WT hPANX1 and its two phosphor-ablating mutants, which is unexpected if these antibodies are indeed specific for phosphorylated PANX1.

Our hPANX1 construct contains a GFP tag at its very C-terminus, which is unlikely to affect PANX1 phosphorylation due to the flexibility of the C-terminus and its considerable distance from the proposed phosphorylation sites. Nevertheless, to thoroughly address this concern, we co-expressed a GFP-free version of hPANX1 with the constitutively active mSrc-Y529F, and performed western blot analysis. Our result showed that both anti-PANX1-pY198 and anti-PANX1-pY308 failed to detect hPANX1, regardless of the presence of the C-terminal GFP tag (*Figure 2—figure supplement 1*, lanes 3–6).

## Anti-PANX1-pY198 and anti-PANX1-pY308 antibodies do not recognize phosphorylated human PANX1 in vitro

We next tested whether anti-PANX1-pY198 and anti-PANX1-pY308 antibodies could detect PANX1 phosphorylation in vitro, as demonstrated in the literature (*DeLalio et al., 2019*), using purified hPANX1 produced in-house and purified human Src with a C-terminal GST tag (Src-GST) purchased from a commercial source.

We first wanted to know whether purified hPANX1 could be phosphorylated by Src in an in vitro context. To this end, we used a pan-specific phosphotyrosine antibody, anti-pY100, which is expected to recognize a wide range of phosphorylated tyrosine sites. Our result clearly showed that anti-pY100 detected only Src (*Figure 3A*, lanes 7–12), which is known to be phosphorylated (*Nada et al., 1991*), but not PANX1 (*Figure 3A*, lanes 1–9), indicating that PANX1 is unlikely to be phosphorylated in our in vitro setting. Similar to anti-pY100, anti-PANX1-pY198 yielded a single band of the same size as the Src protein only in the presence of Src (*Figure 3A*, lanes 10–15). These data confirmed that anti-PANX1-pY198 recognizes only Src, but not PANX1.

Notably, anti-PANX1-pY308 appeared to detect both Src and hPANX1 (*Figure 3A*, lanes 16–18). However, the hPANX1 band was present even in the absence of Src (*Figure 3A*, lane 17), and was not detected by anti-pY100 (*Figure 3A*, lane 7–9). This led us to speculate that anti-PANX1-pY308 may recognize non-phosphorylated PANX1. To test this hypothesis, we performed in vitro phosphorylation experiments on hPANX1 mutants around Y309, including L306A, K307A, V308A, Y309F/A, E310A, I311A, and L312A (*Figure 1B*; *Figure 3B and C*). Our results showed that these mutations also abolished the detection of the hPANX1 band by anti-PANX1-pY308 (*Figure 3B and C*). Thus, our data indicate that the anti-PANX1-pY308 antibody recognizes a non-phosphorylated peptide epitope around Y309 of hPANX1.

We also speculate that anti-PANX1-pY308 prefers Src over hPANX1 as an epitope, because it recognized Src both in its purified form and in crude whole-cell lysate, whereas it only recognized hPANX1 in its purified form, but not in the whole-cell lysate (*Figure 2B*; *Figure 3A–C*). As a result, the Src signal appeared to dominate in the western blot using whole-cell lysate (*Figure 2B*). To test this hypothesis, we blotted whole-cell lysates from HEK293T cells expressing either the WT PANX1 or the Y308F variant. Anti-PANX1-pY308 indeed detect a weak band for WT PANX1, but not for the Y308F variant (*Figure 3D*).

Taken together, by using both cultured HEK293T cells and in vitro phosphorylation experiments, we demonstrated that the anti-PANX1-pY198 and anti-PANX1-pY308 antibodies are non-specific and may not be suitable for studying endogenous PANX1 protein phosphorylation.

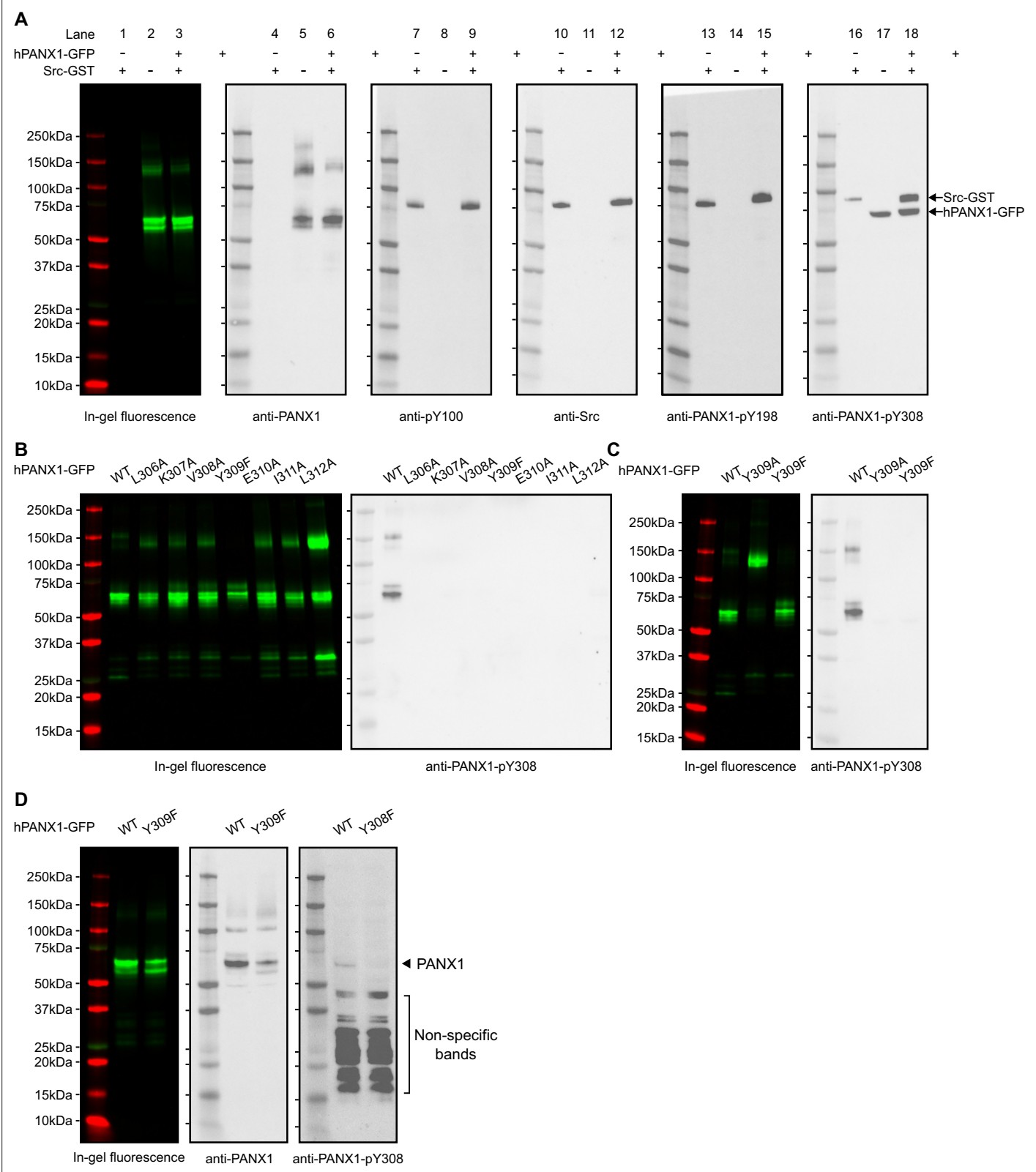

**Figure 3.** The anti-PANX1-pY198 and anti-PANX1-pY308 are not specific to human PANX1 (hPANX1) in vitro. (**A**) In vitro phosphorylation of hPANX1-GFP wild-type (WT) by active human Src-GST protein. The leftmost panel showed the in-gel fluorescence of GFP overlaid with protein marker. The other panels (from left to right) represent the western blot signal of anti-PANX1, anti-pY100, anti-Src, anti-PANX1-PANX1-pY198, and anti-PANX1-pY308 antibodies, respectively. (**B**) In vitro phosphorylation of hPANX1-GFP WT, L306A, K307A, V308A, Y309F, E310A, I311A, or L312A mutants. The left panel

*Figure 3 continued on next page*

Figure 3 continued

showed the in-gel fluorescence of GFP overlaid with protein marker. The right panel showed the western blot signal of anti-PANX1-pY308 antibody. Only hPANX1 WT lane showed western blot signal. (**C**) In vitro phosphorylation of hPANX1-GFP WT, Y309A, or Y309F mutants. The left panel showed the in-gel fluorescence of GFP overlaid with protein marker (red). The right panel showed the western blot signal of anti-PANX1-pY308 antibody. The Y309A mutant rendered PANX1 prone to oligomerization in our SDS gel experiment, but not for Y309F. (**D**) Whole cell lysate of HEK293T cells transiently transfected with hPANX1 WT or Y309F. Left panel showed the in-gel fluorescence of GFP overlaid with protein marker. Middle and right panels represent western blot result using anti-PANX1 and anti-PANX1-pY308 antibodies, respectively. The location where PANX1 is expected to migrate in the SDS-PAGE gel is indicated. Of note, visualizing such a band requires a very long exposure, which resulted in a large number of non-specific bands.

The online version of this article includes the following source data for figure 3:

**Source data 1.** Uncropped immunoblots for *Figure 3*.

**Source data 2.** Raw image for in-gel fluorescence (GFP and mCherry) in *Figure 3A*.

**Source data 3.** Raw image for anti-PANX1 immunoblot in *Figure 3A*.

**Source data 4.** Raw image for anti-pY100 immunoblot in *Figure 3A*.

**Source data 5.** Raw image for anti-Src immunoblot in *Figure 3A*.

**Source data 6.** Raw image for anti-PANX1-pY198 immunoblot in *Figure 3A*.

**Source data 7.** Raw image for anti-PANX1-pY308 immunoblot in *Figure 3A*.

**Source data 8.** Raw image for in-gel fluorescence (GFP and mCherry) in *Figure 3B and C*.

**Source data 9.** Raw image for anti-PANX1-pY308 immunoblot in *Figure 3B and C*.

**Source data 10.** Raw image for in-gel fluorescence (GFP and mCherry) in *Figure 3D*.

**Source data 11.** Raw image for anti-PANX1 immunoblot in *Figure 3D*.

**Source data 12.** Raw image for anti-PANX1-pY308 immunoblot in *Figure 3D*.

## Human PANX1 is not phosphorylated by mouse Src in HEK293T cells

Because commercially available phospho-specific antibodies failed to effectively recognize PANX1, we decided to use the Phos-tag gel to gauge the phosphorylation status of PANX1. The Phos-tag gel allowed for an effective separation of proteins based on their phosphorylation level (*Nagy et al., 2018*; *Nishioka et al., 2021*). We transiently transfected plasmids encoding hPANX1 and mSrc genes into HEK293T cells and analyzed the cell lysates by Phos-tag gel and western blot. *Figure 4A* clearly showed differential migration patterns of mSrc WT, Y529F, and K297M in Phos-tag gel. De-phosphorylating of mSrc by lambda protein phosphatase ($\lambda$-PPase) could partially convert the phosphorylated species into the non-phosphorylated protein (*Figure 4—figure supplement 1*), suggesting that Y529F and K297M mutation could affect the autophosphorylation of mSrc kinase itself (*Figure 4A*, lanes 3–5). The hPANX1, when expressed by itself, showed one prominent band in both Phos-tag gel and SDS-PAGE gel (*Figures 4A and 3B*, lane 2), which likely represents the non-phosphorylated hPANX1 due to the absence of mSrc. We next co-expressed WT hPANX1 with the three mSrc variants, and analyzed the cell lysates with Phos-tag gel to separate phosphorylated and unphosphorylated hPANX1. Contrary to our expectation, the GFP fluorescence only showed a monomeric band irrespective of mSrc activity, suggesting that hPANX1 is insensitive to phosphorylation by mSrc (*Figure 4A*, lanes 6–8 vs 2). To ensure that the expression levels of hPANX1 and mSrc were each comparable across the experimental conditions, we also analyzed the same cell lysate samples using regular SDS-PAGE gel followed by western blot and in-gel fluorescence and found no significant difference in protein expression (*Figure 4B*, lanes 2–8). Our PANX1 gene is tagged with GFP at its C-terminus. To investigate whether GFP would negatively affect the accessibility of the two tyrosine residues to mSrc, we also co-expressed WT hPANX1 without GFP with mSrc variants in HEK293T cells and analyze the cell lysate by Phos-tag gel and western blot analysis. No significant difference was observed between hPANX1 expressed alone or with mSrc variants in Phos-tag gel (*Figure 4—figure supplement 2*, lane 7–9 vs 3). Therefore, our Phos-tag gel analysis indicates that mSrc has no impact on hPANX1 phosphorylation.

However, the inability to detect phosphorylated hPANX1 by Phos-tag gel analysis may be due to the low level of phosphorylation. To address this possibility, we purified hPANX1 protein expressed alone or co-expressed with the constitutively active mSrc-Y529F variant, and conducted liquid chromatography with tandem mass spectrometry (LC-MS/MS) analysis on the samples, which offers high sensitivity for detecting phosphorylated species. While we unambiguously identified MS/MS

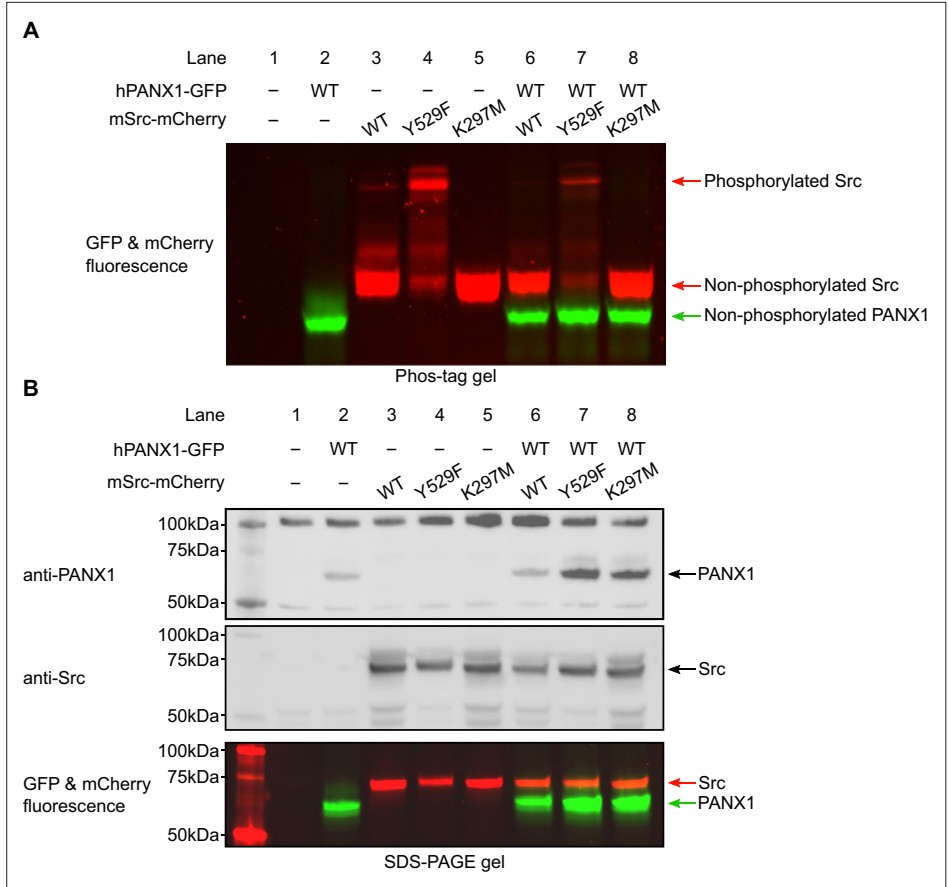

**Figure 4.** Human Pannexin 1 (PANX1) is not phosphorylated by mSrc when expressed in HEK293T cells. (**A**) The Phos-tag gel result of human PANX1 co-expressed with different mSrc mutants. The GFP and mCherry fluorescence signal were overlaid to generate the image. The mSrc kinase of different phosphorylation status are labeled. In all conditions, hPANX1 migrated as a single band. (**B**) Analysis of the same cell lysate sample in B using regular SDS-PAGE gel. Western blot experiment was conducted using anti-PANX1 (top) and anti-Src (middle) antibodies. The anti-PANX1 antibody detected a non-specific band at 100 kDa from the whole-cell lysate with unknown identity. The in-gel fluorescence for GFP and mCherry is shown at the bottom where the position of mSrc and PANX1 protein could be compared with the western blot signal.

The online version of this article includes the following source data and figure supplement(s) for figure 4:

**Source data 1.** Uncropped immunoblots for *Figure 4*.

**Source data 2.** Raw image for in-gel fluorescence (GFP and mCherry) in *Figure 4A*.

**Source data 3.** Raw image for anti-PANX1 immunoblot in *Figure 4B*.

**Source data 4.** Raw image for anti-Src immunoblot in *Figure 4B*.

**Source data 5.** Raw image for in-gel fluorescence (GFP and mCherry) in *Figure 4B*.

**Figure supplement 1.** Dephosphorylation of mSrc protein by lambda protein phosphotase ($\lambda$-PP).

**Figure supplement 1—source data 1.** Uncropped immunoblots for *Figure 4—figure supplement 1*.

**Figure supplement 1—source data 2.** Raw image for in-gel fluorescence (mCherry) in *Figure 4—figure supplement 1* (upper panel).

**Figure supplement 1—source data 3.** Raw image for in-gel fluorescence (mCherry) in *Figure 4—figure supplement 1* (lower panel).

**Figure supplement 2.** Wild-type human PANX1 (hPANX1) without C-terminal GFP tag is not phosphorylated by mSrc.

**Figure supplement 2—source data 1.** Uncropped immunoblots for *Figure 4—figure supplement 2*.

**Figure supplement 2—source data 2.** Raw image for in-gel fluorescence (GFP and mCherry) in *Figure 4—figure supplement 2* (upper panel).

*Figure 4 continued on next page*

eLife Research article

Biochemistry and Chemical Biology

*Figure 4 continued*

**Figure supplement 2—source data 3.** Raw image for anti-PANX1 immunoblot in *Figure 4—figure supplement 2* (middle panel).

**Figure supplement 2—source data 4.** Raw image for anti-PANX1 immunoblot in *Figure 4—figure supplement 2* (lower panel).

**Figure supplement 3.** Coverage map of liquid chromatography with tandem mass spectrometry (LC-MS/MS) analysis of purified human PANX1 (hPANX1)-GFP.

**Figure supplement 4.** Representative liquid chromatography with tandem mass spectrometry (LC-MS/MS) spectra containing the Tyr199 of human PANX1 (hPANX1).

**Figure supplement 5.** Representative liquid chromatography with tandem mass spectrometry (LC-MS/MS) spectrum covering the Tyr309 for human PANX1 (hPANX1).

**Figure supplement 6.** Representative liquid chromatography with tandem mass spectrometry (LC-MS/MS) spectrum covering the Ser385 for human PANX1 (hPANX1).

---

spectra covered both Tyr199 and Tyr309 sites, the tyrosine residues are not phosphorylated even when hPANX1 is co-expressed with mSrc-Y529F (*Figure 4—figure supplements 3–5*). Interestingly, we observed hPANX1 Ser385 phosphorylation when mSrc-Y529F is present (*Figure 4—figure supplement 6*). However, out of 67 MS/MS spectra, only five spectra contain the phosphorylated Ser385 (~7.5%). We note that phosphorylated peptides may display a very different fragmentation and ionization efficiency compared to the non-phosphorylated counterparts (*Dephoure et al., 2013*). Therefore, the ratio of the MS/MS spectra does not represent the true phosphorylation level of hPANX1 protein. Our Phos-tag gel analysis suggests that the Ser385 phosphorylation could be at a very low level as we did not observe other major bands in Phos-tag gel when hPANX1 is co-expressed with mSrc-Y529F (*Figure 4A*). Nevertheless, our result clearly demonstrated that Tyr199 and Tyr309 of hPANX1 are not phosphorylated by mSrc, at least not to a detectable level, when transiently expressed in HEK293T cells.

## Phosphorylation status of human PANX1 is not altered by Src in Neuro2A cells

Since our results conducted in HEK293T cells contradicted to what was known in the literature, we wanted to test whether our result was cell line dependent. To this end, we co-expressed hPANX1 and mSrc in Neuro2A cells, which were previously used as a model system to access the phosphorylation of Y308 by NMDAR activation (*Weilinger et al., 2016*). We noticed that hPANX1 protein expressed in Neuro2A cells was heavily glycosylated as indicated by its heterogeneous migration bands in regular SDS-PAGE gel (*Figure 5A*, lane 2). However, the multimeric bands can be reduced to a single band after PNGase F treatment, confirming that the heterogeneity is indeed due to N-linked glycosylation (*Figure 5A*, lane 3). Next, we tested the phosphorylation status of the samples using Phos-tag gel. Our result showed that deglycosylated hPANX1 migrates to the same location independent of mSrc, suggesting that Src does not alter the phosphorylation status of hPANX1 (*Figure 5B*, lanes 8, 10, 12 vs 3).

## Mouse PANX1 is not phosphorylated by Src in HEK293T cells

To rule out the possibility that hPANX1 may not be recognized by mouse Src, we repeated the transient transfection experiment using mPANX1 (*Figure 6*). Using Phos-tag gel, we failed to observe any difference between mPANX1 expressed alone or co-expressed with mSrc variants in HEK293T cells, indicating that mPANX1 behaves in the same way as hPANX1 and is not phosphorylated by mSrc (*Figure 6A*, lanes 6–8 vs 2). The expression levels of mPANX1 and mSrc are each comparable across different experimental conditions as judged by western blot and in-gel fluorescence image (*Figure 6B*).

We noted that the PANX1 antibody from Abcam (Catalogue number: ab124131) seems to detect other non-specific bands in our experiment, including a major band close to 100 kDa and a minor band at approximately 50 kDa (*Figures 4B and 6B*). To sort out the identity of the two bands, we carried out a de-glycosylation experiment on non-transfected HEK293T cells and hPANX1-expressing cells (*Figure 6—figure supplement 1*). Our result suggests that the 50 kDa band could be partially

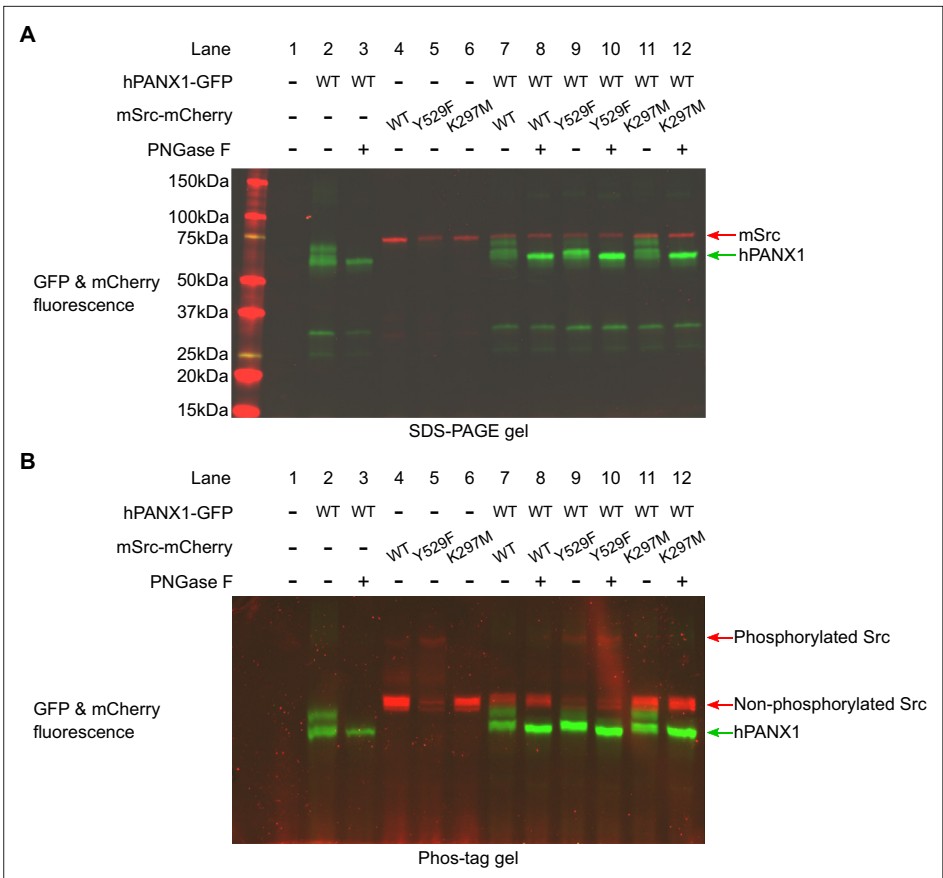

**Figure 5.** Human Pannexin 1 (PANX1) is not phosphorylated by Src when expressed in Neuro2A cells. (**A**) A regular SDS gel showing the GFP and mCherry fluorescence signal of human PANX1 (hPANX1) and mSrc co-expressed in Neuro2A cells. For each condition containing PANX1, PNGase F treatment is performed to de-glycosylate the protein. (**B**) A PhosTag gel result using the same samples analyzed in **A**.

The online version of this article includes the following source data for figure 5:

**Source data 1.** Uncropped immunoblots for *Figure 5*.

**Source data 2.** Raw image for in-gel fluorescence (GFP and mCherry) in *Figure 5A*.

**Source data 3.** Raw image for in-gel fluorescence (GFP and mCherry) in *Figure 5B*.

de-glycosylated and migrate to a location where overexpressed PANX1 protein is expected to stay, whereas the 100 kDa band is insensitive to PNGase F treatment (*Figure 6—figure supplement 1*). Thus, the lower band (50 kDa) likely represents endogenous PANX1 from HEK293T cells, while the 100 kDa band is a non-specific target of anti-PANX1 antibody. This result highlights the caveats/complications of solely relying on western blot to interpret the band identity. When combined with the fluorescence signal, it is possible to determine with more confidence which bands correspond to the full-length mPANX1 and mSrc protein in the SDS-PAGE gel.

## The PANX1 current remains unchanged in the presence of active mSrc

To investigate whether hPANX1 channel activity is affected by the presence of active mSrc, we co-expressed GFP-tagged hPANX1 with the mCherry-tagged constitutively active mSrc-Y529F, and performed whole-cell patch clamp analysis. We targeted cells showing both green and red fluorescence, indicative of expression of both hPANX1 and mSrc proteins (*Figure 7A*). We did not observe any significant difference in the CBX-sensitive currents compared to the currents when hPANX1 was expressed alone (*Figure 7B*). This data strongly suggests that mSrc does not affect the gating of hPANX1 channel in HEK293T cells, further reinforcing the notion that PANX1 is not phosphorylated by Src.

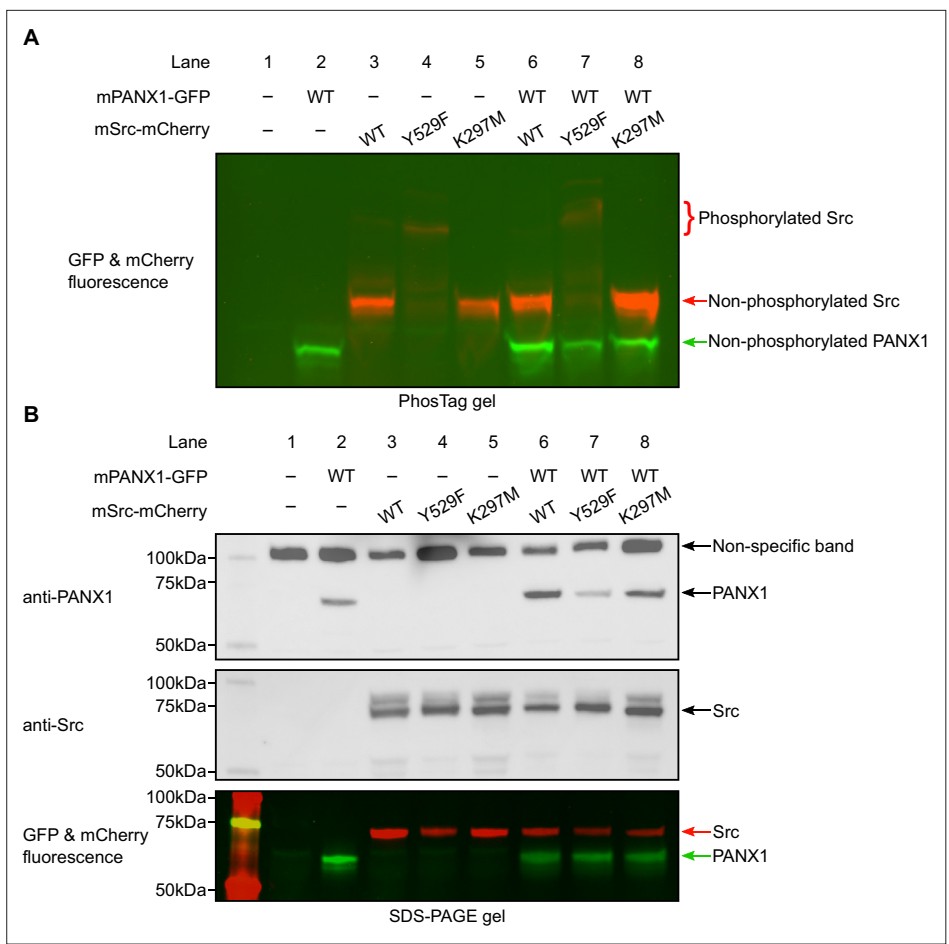

**Figure 6.** Mouse Pannexin 1 (PANX1) is not phosphorylated by Src when expressed in HEK293T cells. (**A**) A Phos-tag gel result of mPANX1 co-expressed with different mSrc mutants. The GFP and mCherry fluorescence signals are overlaid to generate the image. The mSrc kinase of different phosphorylation status are labeled. In all the conditions, mouse PANX1 migrates to a single band. (**B**) Analysis of the same cell lysate sample in B using regular SDS-PAGE gel. Western blot experiment is conducted using anti-PANX1 (top) and anti-Src (middle) antibodies. The in-gel fluorescence for GFP and mCherry is shown at the bottom where the position of mSrc and mPANX1 protein could be compared with the western blot signal.

The online version of this article includes the following source data and figure supplement(s) for figure 6:

**Source data 1.** Uncropped immunoblots for *Figure 6*.

**Source data 2.** Raw image for in-gel fluorescence (GFP and mCherry) in *Figure 6A*.

**Source data 3.** Raw image for anti-PANX1 immunoblot in *Figure 6B*.

**Source data 4.** Raw image for anti-Src immunoblot in *Figure 6B*.

**Source data 5.** Raw image for in-gel fluorescence (GFP and mCherry) in *Figure 6B*.

**Figure supplement 1.** The anti-Pannexin 1 (PANX1) antibody detects both human PANX1 (hPANX1) and non-specific proteins.

**Figure supplement 1—source data 1.** Uncropped immunoblots for *Figure 6—figure supplement 1*.

**Figure supplement 1—source data 2.** Raw image for anti-PANX1 immunoblot in *Figure 6—figure supplement 1* (left panel).

**Figure supplement 1—source data 3.** Raw image for anti-PANX1 immunoblot in *Figure 6—figure supplement 1* (right panel).

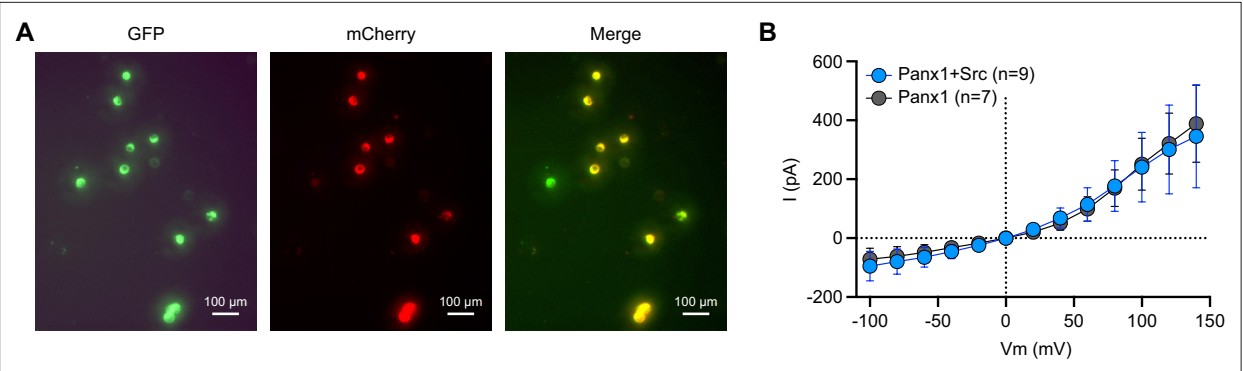

**Figure 7.** The Pannexin 1 (PANX1) current remains unchanged in the presence of active mSrc. (**A**) Representative fluorescent images of HEK293T cells transfected with human PANX1 hPANX1-GFP and mSrc-Y529F-mCherry. Cells that show both green (GFP) and red (mCherry) fluorescence are selected for whole-cell patch-clamp analysis. (**B**) The CBX sensitive whole-cell current of hPANX1-GFP and hPANX1 +mSrc-Y529F-mCherry. Two-way ANOVA was conducted to compare the hPANX1-GFP and hPANX1+mSrc-Y529F-mCherry. The analysis revealed a non-significant difference between the two groups $(F_{(1, 182)}=0.07286, p=0.7875)$. Data are presented as mean ± SEM. The number of cells (n) measured are indicated.

The online version of this article includes the following source data for figure 7:

**Source data 1.** The raw fluorescent images of the panels are shown in *Figure 7A*.

**Source data 2.** The raw data plotted in *Figure 7B*.

## Discussion

Protein phosphorylation is a common mechanism to regulate the physiology of large-pore channel function. For example, Cx43 phosphorylation by p34$^{cdc2}$ kinase is associated with channel internalization during mitosis (*Xie et al., 1997*; *Kanemitsu et al., 1998*; *Lampe et al., 1998*). The volume-regulated anion channel was also recently found to be phosphorylated by MSK1 kinase, which in turn controls cellular chloride efflux (*Serra et al., 2021*). Not unexpectedly, PANX1 phosphorylation is also implicated in several physiological conditions. The common scheme of these processes is that PANX1 acts downstream of a membrane receptor, including P2X7, NMDARs, α7 nAChR, TNFR1, and α1-AR, which senses various extracellular signal and results in PANX1 activation via a yet uncharacterized intracellular signaling cascade (*Locovei et al., 2007*; *Maldifassi et al., 2021*; *Weilinger et al., 2012*; *Weilinger et al., 2016*; *Lohman et al., 2015*; *DeLalio et al., 2019*). The Src non-receptor tyrosine kinase appears to be a key player during this process, as inhibiting Src activity by kinase inhibitors could quench PANX1 opening (*Locovei et al., 2007*; *Maldifassi et al., 2021*; *Weilinger et al., 2012*; *Lohman et al., 2015*; *DeLalio et al., 2019*). Several evidence to support the notion that Src could directly phosphorylate PANX1 comes from western blot analysis, in which phosphor-tyrosine antibodies against Y198 and Y308 of murine PANX1 were used. In the study by *DeLalio et al., 2019*, attempts have been made to compare the effectiveness of pY198 antibody against WT mPANX1 and Y198F mutant in HEK293T cells. However, the signal by the pY198 antibody does not seem to match where the mPANX1 protein is expected to appear, although the blot is rather noisy (see *Figure 2* of *DeLalio et al., 2019*). In the study by Weilinger et al., pY308 signal is present even for Y308F mutant, suggesting that the antibody may not be specific for pY308 of rat PANX1 (see Figure 5b of *Weilinger et al., 2016*). Unfortunately, these antibodies are taken for granted and continued to be used for scientific research (*Metz and Elvers, 2022*).

Here, we combined multiple techniques, including Phos-tag gel, western blot, LC-MS/MS, and patch-clamp electrophysiology to investigate Src-dependent PANX1 phosphorylation. Our results, in contrast to previous findings, indicate that Tyr199 and Tyr309 of hPANX1 are not effective substrates for Src kinase, at least in HEK293T and Neuro2A cells or in in vitro experiments using purified hPANX1. Nevertheless, it is still possible that these two tyrosine sites are phosphorylated by other tyrosine kinases in other cell lines. For example, recent electrospray ionization mass spectrometry analysis on hPANX1 identified Tyr150 as a minor phosphorylation site in human triple-negative breast cancer Hs578T cells (*Nouri-Nejad et al., 2021*). Our mass spectrometry data failed to detect any fragments corresponding to a phosphorylated Tyr150 even when an active mSrc kinase is coexpressed with hPANX1, suggesting that Tyr150 may not be effectively recognized by mSrc. Consistent with this, a

number of other tyrosine kinases have been predicted as the candidates for Tyr150 phosphorylation (*Nouri-Nejad et al., 2021*). Likewise, our study also does not rule out the possibility that Src may still be involved in the regulation of PANX1, given the fact that Src kinase inhibitors affect PANX1 channel opening under these conditions (*Locovei et al., 2007*; *Maldifassi et al., 2021*; *Weilinger et al., 2012*; *Lohman et al., 2015*; *DeLalio et al., 2019*). It's possible that additional proteins are involved to relay the signaling output to PANX1. For example, the endoplasmic reticulum (ER)-resident stromal interaction molecules (STIM1/2) have recently been noted to mediate events between NMDAR and PANX1(*Patil et al., 2022*). Future studies are required to fully decipher the underlying mechanisms.

Given the fact that the commercially available antibodies against phospho-PANX1 are non-specific, we encourage the PANX1 research community to re-examine results derived from the Src-mediated PANX1 phosphorylation and the associated antibodies. The technological development in the production of phospho-specific antibodies represents a breakthrough in the field of protein kinase and phosphoprotein research, providing unprecedented detection precision and accuracy. While phospho-specific antibodies have been successfully developed for many protein targets, great caution still needs to be taken when analyzing novel phosphorylation sites, and control experiments (such as the use of phosphor-ablating mutants and pan-specific phosphorylation antibodies) should be conducted. Ideally, other complementary techniques such as mass spectrometry and Phos-tag gel analysis should be sought to independently verify the conclusion.

## Materials and methods

### Plasmid constructs

Full-length human PANX1 in the pEGC Bacmam vector from our previous study was used (*Ruan et al., 2020*). Mouse PANX1 (UniprotID: Q9JIP4) was synthesized by GenScript and subcloned into the pEGC Bacmam vector (*Goehring et al., 2014*). The translated product contains the human or mouse PANX1 protein, a thrombin digestion site (LVPRGS), an enhanced GFP protein, and an 8 x His tag. pCMV5 mouse Src was a gift from Joan Brugge & Peter Howley (Addgene plasmid # 13663; http://n2t.net/addgene:13663; RRID:Addgene_13663). The mouse Src gene was subcloned into the pEGN Bacmam vector (*Goehring et al., 2014*). The translated product contains an mCherry protein, a thrombin digestion site, and the Src protein. Primers for site-directed mutagenesis were designed using QuikChange Primer Design website (https://www.agilent.com/store/primerDesignProgram.jsp) and synthesized by Eurofins Genomics. The QuikChange mutagenesis protocol was used to generate all the mutants of the study. Sanger sequencing was performed to identify positive clones.

### Cell lines

HEK293T cells are purchased from Sigma-Aldrich (Catalogue Number: 96121229). Neuro2A cells are purchased from ATCC (Catalogue Number: CCL-131). The cells are authenticated and are myco-plasma contamination tested negative by the vendor.

### Cell culture

Adherent HEK293T cells were grown in DMEM media supplemented with 10% fetal bovine serum. Transient transfection was conducted using lipofectamine-2000 by following the manufacturer's protocol. Specifically, the cells were cultured in 60 mm Petri dishes until 80% confluency. Transfection solution was made by mixing 500 ng of plasmid DNA, 4 uL of lipofectamine-2000 reagent, and 100 uL Opti-MEM media. After 10 min incubation at room temperature, the DNA-lipid complexes were added to the cell culture and incubated at 37 °C. The next day, 10 mM sodium butyrate was added to the cells to boost protein expression. The cell culture was then grown at 30 °C for another day before harvesting. The cell pellet was flash-frozen with liquid nitrogen and stored at –80 °C. For co-transfection experiments, equal amounts of hPANX1 (250 ng) and mSrc (250 ng) plasmids were used in the transfection mixture. For Neuro2A cell culture, EMEM media is used instead of DMEM. All the other procedures described above are the same for expressing proteins in Neuro2A cells.

### Western blot

For the western blot experiment, the cell pellet was lysed in TBS buffer (20 mM Tris pH 8.0, 150 mM NaCl) with protease an inhibitor cocktail (1 mM phenylmethylsulfonyl fluoride, 2 mM pepstatin, 0.8 μM

aprotinin and 2 µg/ml leupeptin), 1 mM sodium orthovanadate and 1% glyco-diosgenin detergent for 30 min on ice. The lysate was clarified by centrifugation at 21,000 rpm for 20 min and the soluble portions were mixed with 2 X SDS loading buffer supplemented with 5% 2-Mercaptoethanol. The samples were resolved in precast gradient Tris-Glycine gel (4–20%) or 7.5% PhosTag gel. A Chemidoc instrument was used to directly image the in-gel fluorescence signal after electrophoresis. Subsequently, the protein in the gel was transferred to the PVDF membrane using the semi-dry transfer buffer (48 mM Tris base, 39 mM glycine, 20% methanol). The membrane is blocked in the TBST buffer (20 mM Tris pH8.0, 150 mM NaCl, 0.1% Tween 80) with 4% non-fat milk for 1 h at room temperature. Afterward, primary antibodies (1:2000 dilution) were incubated with the membrane overnight at 4 °C. The next day, the membrane was washed with TBST buffer for four times, 10 min each before goat anti-rabbit/mouse IgG secondary antibodies were added (1:25000 dilution). After 1 hr, the secondary antibodies were discarded and the membrane was washed again with TBST for four times, 10 min each. The western blot signal was detected using ECL Pierce substrate and imaged using a Chemidoc instrument. A brightfield image was overlaid with the illuminance signal to visualize the position of protein markers in the membrane. The anti-PANX1 antibody is obtained from Abcam (Catalogue number: ab124131). The anti-Src (Catalogue number: 36D10) and anti-pY100 (Catalogue number: 9411) antibodies are obtained from Cell Signaling. The anti-PANX1-pY198 (Catalogue number: ABN1681) and anti-PANX1-pY308 (Catalogue number: ABN1680) antibodies were obtained from Millipore Sigma.

## De-glycosylation assay

For de-glycosylation experiment, Neuro2A/HEK293T cells expressing designated genes were solubilized in TBS buffer with protease inhibitor cocktail (1 mM phenylmethylsulfonyl fluoride, 2 mM pepstatin, 0.8 µM aprotinin and 2 µg/ml leupeptin), 1 mM sodium orthovanadate and 1% glyco-diosgenin detergent for 30 min on ice. The samples were clarified by centrifugation at 21,000 rpm for 30 min. The de-glycosylation reaction was made by mixing 16 uL of the supernatant with 2 uL of PNGase F enzyme and 2 uL of GlycoBuffer 2 (10X). The control reaction replaced the PNGase F enzyme with water. The reaction was allowed to occur at room temperature overnight. The next day, the samples were mixed with 20 uL 2X SDS sample-loading buffer (BioRad) supplemented with 5% βME and resolved by SDS-PAGE. The gel was imaged in the ChemiDoc system by probing the GFP and mCherry fluorescence signal.

## De-phosphorylation assay

For the dephosphorylation experiment, mSrc-mCherry WT, Y529F, and K297M were expressed in HEK293T cells. The cells were lysed in TBS buffer (20 mM Tris pH8, 150 mM NaCl) with 1% glyco-diosgenin detergent for 30 min on ice. After clarifying the non-soluble debris by centrifugation at 21,000 rpm for 20 min, 20 µL supernatant was mixed with 2.5 µL of 10 X NEBuffer for Protein Metal-loPhosphatases, 2.5 µL 10 mM $MnCl_2$ and 2 µL Lambda Protein Phosphatase (NEB). The dephosphorylation reaction was allowed to occur at 30 °C overnight. Control samples without adding Lambda Protein Phosphatase are placed on ice. The next day, the samples were mixed with 2X SDS sample-loading buffer (BioRad) supplemented with 5% βME and resolved by SDS-PAGE or Phos-tag gel. The gel was imaged in the ChemiDoc system by probing the mCherry fluorescence signal or western blot analysis.

## Protein purification

For in vitro phosphorylation and mass-spectrometry analysis, the hPANX1 WT or mutants were expressed alone or co-expressed with mSrc Y529F mutant in HEK293T cells. Specifically, 50 µg of plasmid DNA (in the case of the co-expressing experiment, 25 µg of human PANX1 plasmid and 25 µg of mouse Src Y529F mutant plasmid was used) was mixed with 150 ug of PEI 25 K (Polysciences) in a and incubated at room temperature for 30 min. The DNA-PEI complex was then added to the suspension cell culture of HEK293T cells at a density of $2\times10^6$ cells/ml in FreeStyle medium (Gibco). After growing at 37 °C for 8 hr, 10 mM sodium butyrate was added, and the temperature was changed to 30 °C for 40 hr. The cells were then harvested and stored at –80 °C until purification. The protein purification procedure was described in our previous study (*Ruan et al., 2020*).

## In vitro phosphorylation assay

The in vitro phosphorylation is conducted using the human Src-GST protein (Sigma-Aldrich, Catalogue number: S1076) according to the manufacturer's protocol. Specifically, 1.5 µg of purified PANX1 protein is mixed with kinase assay buffer (25 mm MOPS, pH 7.2, 20 mM $MgCl_2$, 12.5 mM $MnCl_2$, 5 mM EGTA, 2 mM EDTA, 0.25 mM DTT), diluted 1:5 with 50 ng/µl bovine serum albumin (BSA). The reaction was supplemented with 0.25 mM ATP, and 0.3 µg recombinant human (active) Src-GST kinase. Samples were incubated for 1 h at 30 °C using a PCR thermocycler. After 1 hr, 2 x SDS loading buffer (BioRad) is mixed with the sample to stop the reaction. The sample is then resolved by an Tris-Glycine gel for subsequent western blot analysis.

## LC-MS/MS

Purified hPANX1 protein w- or w/o- co-expressing mSrc Y529F mutant was resolved in SDS-PAGE gel and the band corresponding to hPANX1 protein was cut and subjected to in-gel digestion with trypsin. Half of each digested sample was analyzed by nano LC-MS/MS with a Waters M-Class HPLC system interfaced with a Thermo Fisher Fusion Lumos mass spectrometer. Peptides were loaded on a trapping column and eluted over a 75 µm analytical column at 350 nL/min; both columns were packed with Luna C18 resin (Phenomenex). The mass spectrometer was operated in data-dependent mode, with the Orbitrap operating at 60,000 FWHM and 15,000 FWHM for MS and MS/MS, respectively. The instrument was run with a 3 s cycle for MS and MS/MS.

Data were searched using a local copy of Mascot (Matrix Science) with the following parameters. Enzyme: Trypsin/P; Database: SwissProt Human (concatenated forward and reverse plus common contaminants); Fixed modification: Carbamidomethyl (C) Variable modifications: Oxidation (M), Acetyl (N-term), Pyro-Glu (N-term Q), Deamidation (N/Q); Mass values: Monoisotopic; Peptide Mass Tolerance: 10 ppm; Fragment Mass Tolerance: 0.02 Da; Max Missed Cleavages: 2. Mascot DAT files were parsed into Scaffold (Proteome Software) for validation, filtering and to create a non-redundant list per sample. Data were filtered using 1% protein and peptide FDR and requiring at least two unique peptides per protein.

## Electrophysiology

HEK293T cells were transfected using Lipofectamine 2000 (Thermo Fisher) according to the manufacturer's protocol. The transfected cells were incubated at 37 °C for 18–24 hr before electrophysiological measurements. Whole-cell recordings were performed using a Multiclamp 700B (Axon Instruments) and Clampex software, with pipettes of 3–5 MOhm resistance filled with an internal solution containing 145 mM NaCl, 10 mM Hepes, 10 mM EGTA, pH adjusted to 7.4. The external bath solution contained 160 mM NaCl, 10 mM Hepes, 3 mM KCl, 2 mM $CaCl_2$, and 1 mM $MgCl_2$, also adjusted to pH 7.4. During the recordings, voltage steps ranging from −100 mV to +140 mV were applied, each lasting 100ms with a 20 mV increment between steps, and membrane currents were digitally recorded at 10 kHz and filtered at 2 kHz. To precisely measure the PANX1 channel's current, carbenoxolone disodium salt (CBX), a blocker of the channel, was added to the bath solution at a final concentration of 0.1 mM, and CBX-sensitive currents were subsequently calculated by comparing the difference in current amplitude in a cell with and without the presence of CBX.

## Acknowledgements

We acknowledge Dr. Henriette Remmer at the Proteomics & Peptide Synthesis Core in the University of Michigan for the helpful discussion with Mass spectrometry data. We thank members of the Du & Lü Labs for thoughtful discussions. ZR is supported by an American Heart Association (AHA) postdoctoral fellowship (grant 20POST35120556) and the National Institute of Health (NIH) (grant K99NS128258). WL is supported by the NIH (grant R35GM138321). JD is supported by a McKnight Scholar Award, a Klingenstein-Simon Scholar Award, a Sloan Research Fellowship in Neuroscience, a Pew Scholar in the Biomedical Sciences award, and an NIH grant (R01NS111031).

## Additional information

### Funding

| Funder | Grant reference number | Author |
| --- | --- | --- |
| American Heart Association | 20POST35120556 | Zheng Ruan |
| National Institute of Neurological Disorders and Stroke | K99NS128258 | Zheng Ruan |
| National Institute of General Medical Sciences | R35GM138321 | Wei Lü |
| National Institute of Neurological Disorders and Stroke | R01NS111031 | Juan Du |
| McKnight Endowment Fund for Neuroscience | | Juan Du |
| Esther A. and Joseph Klingenstein Fund | | Juan Du |
| Pew Charitable Trusts | | Juan Du |
| Alfred P. Sloan Foundation | | Juan Du |

The funders had no role in study design, data collection and interpretation, or the decision to submit the work for publication.

### Author contributions

Zheng Ruan, Data curation, Validation, Investigation, Visualization, Writing – original draft, Writing – review and editing; Junuk Lee, Data curation, Validation, Investigation, Visualization, Writing – review and editing; Yangyang Li, Data curation, Investigation; Juan Du, Wei Lü, Conceptualization, Supervision, Funding acquisition, Writing – review and editing

### Author ORCIDs

Zheng Ruan (ID) https://orcid.org/0000-0002-4412-4916
Junuk Lee (ID) http://orcid.org/0000-0002-3596-2651
Juan Du (ID) http://orcid.org/0000-0003-1467-1203
Wei Lü (ID) http://orcid.org/0000-0002-3009-1025

Reviewer #1 (Public Review): https://doi.org/10.7554/eLife.95118.3.sa1
Reviewer #2 (Public Review): https://doi.org/10.7554/eLife.95118.3.sa2
Reviewer #3 (Public Review): https://doi.org/10.7554/eLife.95118.3.sa3
Author response https://doi.org/10.7554/eLife.95118.3.sa4

## Additional files

### Supplementary files
• MDAR checklist

### Data availability

The LC-MS/MS data for WT human PANX1 with or without co-expressing Src-Y529F mutant is deposited in Dryad database under https://doi.org/10.5061/dryad.4tmpg4fh7. The raw images of the gels/immunoblots are provided as the source data for the corresponding figures.

The following dataset was generated:

| Author(s) | Year | Dataset title | Dataset URL | Database and Identifier |
|---|---|---|---|---|
| Ruan Z, Lü W, Du J | 2024 | LC-MS/MS data for WT human PANX1 with or without co-expressing Src-Y529F mutant | https://doi.org/10.5061/dryad.4tmpg4fh7 | Dryad Digital Repository, 10.5061/dryad.4tmpg4fh7 |

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
