## [Editor Report · eLife assessment]

The current manuscript re-examines an established claim in the literature that human PANX-1 is regulated by Src kinase phosphorylation at two tyrosine residues, Y199 and Y309. This issue is **important** for our understanding of Pannexin channel regulation. The authors present an extensive series of experiments that fail to detect PANX-1 phosphorylation at these sites. Although the authors' approach is more rigorous than the previous studies, this work relies primarily on negative results that are not unambiguously definitive; the work nonetheless provides a **compelling** reason for the field to reexamine conclusions drawn in earlier studies.

---

## [Referee Report · Reviewer #1 (Public Review)]

The current manuscript revisits previous reports in the literature. The human Pannexin 1 channel is regulated by phosphorylation at two residues by Src kinase. From this series of experiments, the authors conclude that PANX-1 is not phosphorylated at these residues.

The biggest strength of the manuscript is the comprehensiveness of the approach. The authors recapitulate prior experiments in the literature and also add a series of new, orthogonal experiments that all examine the claim of PANX-1 phosphorylation. The breadth of the reported experiments extends over multiple cell lines and protein constructs, in vitro purified proteins, mass spec, different phosphorylation detection reagents and antibodies, and functional electrophysiology assays that show that the addition of Src does not impact gating. The combined weight of all these data strongly suggests that the field should re-examine the claim that PANX-1 is regulated by phosphorylation at Y199 and Y309.

Another strength is that the authors go beyond simply showing that the antibodies do not recognize phosphorylated PANX-1. They also provide potential mechanisms for how the antibodies may be misleading. Both antibodies recognize phosphorylated Src-1. In the case of anti-PANX1-pY308, the authors provide solid mutagenesis evidence that the antibody also weakly recognizes a non-phosphorylated epitope of PANX1 in the same region as the tyrosine. This helps make a convincing case.

Such experiments, while not glamorous, have great practical importance for developing an accurate understanding of how Pannexin channels are regulated.

---

## [Referee Report · Reviewer #2 (Public Review)]

The widely distributed pannexin 1 (PANX1) is an ATP-permeable channel that plays an important role in intercellular communication and has been implicated in various pathophysiological processes and diseases. Previous studies have demonstrated that PANX1 can be phosphorylated at two molecular sites via the non-receptor kinase Src, thereby leading to channel opening and ATP release. In this paper, the authors used a variety of methods to detect tyrosine phosphorylation modification of PANX1 channel protein, however, their results showed that commercially available antibodies against the two phosphorylation sites used in previous studies did not work well, in other words, phosphorylation changes in PANX1 could not be detected by those antibodies. Therefore, the authors call for the re-examination and evaluation of previous research results.

In general, this is a meticulous study, using different detection methods and different expression systems.

---

## [Referee Report · Reviewer #3 (Public Review)]

The manuscript by Ruan et al. addresses an important issue in Panx1 research, i.e. the activation of the channel formed by Panx1 via protein phosphorylation. If the authors' conclusions are correct, the previous claims for Panx1 phosphorylation on the basis of the commercial anti-phospho-Panx1 antibodies would be in question.

This is a very detailed and comprehensive analysis making use of state-of-the-art techniques, including mass spectrometry and phos-tag gel electrophoresis.

In general, the study is well-controlled as relating to negative controls.

The value of this manuscript is, that it could spawn new, more function-oriented studies on the activation of Panx1 channels.

The weaknesses identified previously are reproduced below:

Weaknesses:

Although the manuscript addresses an important issue, the activation of the ATP-release channel Panx1 by protein phosphorylation, the data provided do not support the firm conclusion that such activation does not exist. The failure to reproduce published data obtained with commercial anti-phospho Panx1 antibodies can only be of limited interest for a subfield.

(1) The title claiming that "Panx1 is NOT phosphorylated..." is not justified by the failure to reproduce previously published data obtained with these antibodies. If, as claimed, the antibodies do not recognize Panx1, their failure cannot be used to exclude tyrosine phosphorylation of the Panx1 protein. There is no positive control for the antibodies.

(2) The authors claim that exogenous SRC expression does not phosphorylate Y198. DeLalio et al. 2019 show that Panx1 is constitutively phosphorylated at Y198, so an effect of exogenous SRC expression is not necessarily expected.

(3) The authors argue that the GFP tag of Panx1at the COOH terminus does not interfere with folding since the COOH modified (thrombin cleavage site) Panx1 folds properly, forming an amorphous glob in the cryo-EM structure. However, they do not show that the COOH-modified Panx1 folds properly. It may not, because functional data strongly suggest that the terminal cysteine dives deep into the pore. For example, the terminal cysteine, C426, can form a disulfide bond with an engineered cysteine at position F54 (Sandilos et al. 2012).

(4) The authors dismiss the additional arguments for tyrosine phosphorylation of Panx1 given by the various previous studies on Panx1 phosphorylation. These studies did not, as implied, solely rely on the commercial anti-phospho-Panx1 antibodies, but also presented a wealth of independent supporting data. Contrary to the authors' assertion, in the previous papers the pY198 and pY308 antibodies recognized two protein bands in the size range of glycosylated and partial glycosylated Panx1.

(5) A phosphorylation step triggering channel activity of Panx1 would be expected to occur exclusively on proteins embedded in the plasma membrane. The membrane-bound fraction is small in relation to the total protein, which is particularly true for exogenously expressed proteins. Thus, any phosphorylated protein may escape detection when total protein is analyzed. Furthermore, to be of functional consequence, only a small fraction of the channels present in the plasma membrane need to be in the open state. Consequently, only a fraction of the Panx1 protein in the plasma membrane may need to be phosphorylated. Even the high resolution of mass spectroscopy may not be sufficient to detect phosphorylated Panx1 in the absence of enrichment processes.

(6) In the electrophysiology experiments described in Figure 7, there is no evidence that the GFP-tagged Panx1 is in the plasma membrane. Instead, the image in Figure 7a shows prominent fluorescence in the cytoplasm. In addition, there is no evidence that the CBX-sensitive currents in 7b are mediated by Panx1-GFP and are not endogenous Panx1. Previous literature suggests that the hPanx1 protein needs to be cleaved (Chiu et al. 2014) or mutated at the amino terminus (Michalski et al 2018) to see voltage-activated currents, so it is not clear that the currents represent hPANX1 voltage-activated currents.

Note from the editors: The authors provided a rebuttal to the latest review, but no additional data, so we encourage readers to read the concerns and the author responses.

---

## [Author Response]

The following is the authors’ response to the original reviews.

**Reviewer #3 (Public Review):**
Summary:It has been proposed in the literature, that the ATP release channel Panx1 can be activated in various ways, including by tyrosine phosphorylation of the Panx1 protein. The present study reexamines the commercial antibodies used previously in support of the phosphorylation hypothesis and the presented data indicate that the antibodies may recognize proteins unrelated to Panx1. Consequently, the authors caution about the use and interpretation of results obtained with these antibodies.Strengths:The manuscript by Ruan et al. addresses an important issue in Panx1 research, i.e. the activation of the channel formed by Panx1 via protein phosphorylation. If the authors' conclusions are correct, the previous claims for Panx1 phosphorylation on the basis of the commercial anti-phospho-Panx1 antibodies would be in question.This is a very detailed and comprehensive analysis making use of state-of-the-art techniques, including mass spectrometry and phos-tag gel electrophoresis.In general, the study is well-controlled as relating to negative controls.The value of this manuscript is, that it could spawn new, more function-oriented studies on the activation of Panx1 channels.Weaknesses:Although the manuscript addresses an important issue, the activation of the ATP-release channel Panx1 by protein phosphorylation, the data provided do not support the firm conclusion that such activation does not exist. The failure to reproduce published data obtained with commercial anti-phospho Panx1 antibodies can only be of limited interest for a subfield.(1) The title claiming that "Panx1 is NOT phosphorylated..." is not justified by the failure to reproduce previously published data obtained with these antibodies. If, as claimed, the antibodies do not recognize Panx1, their failure cannot be used to exclude tyrosine phosphorylation of the Panx1 protein. There is no positive control for the antibodies.

The full title of our manuscript is “Human Pannexin 1 Channel is NOT Phosphorylated by Src Tyrosine Kinase at Tyr199 and Tyr309”. The major conclusion of our manuscript shall not be extended to the claim that “Panx1 is NOT phosphorylated”. This is by no means our conclusion. In fact, the LC-MS/MS data from both ours and others have shown that PANX1 is phosphorylated at both serine and tyrosine sites (1). However, we provided solid evidence that Tyr199 and Tyr309 of human PANX1 are not effective substrate of the Src kinase.

We did provide several positive controls for the antibodies in our study. We showed that the anti-PANX1 and anti-Src antibodies unambiguously recognized PANX1 and Src, respectively (Figure 3A), and that a pan-specific phosphotyrosine antibody (P-Tyr-100) unambiguously recognized phosphorylated Src (Figure 3A)—as expected—but did not recognize PANX1. In addition, we demonstrated that the two antibodies in question (anti-PANX1-pY198 and anti-PANX1-pY308) did produce signals in our western blot analysis, but we provided compelling evidence that the bands produced by these antibodies do not correspond to PANX1 (Figure 2B).

(2) The authors claim that exogenous SRC expression does not phosphorylate Y198. DeLalio et al. 2019 show that Panx1 is constitutively phosphorylated at Y198, so an effect of exogenous SRC expression is not necessarily expected.

We have unambiguously identified peptide fragments containing non-phosphorylated Y198 in our LC-MS/MS experiment, none corresponds to a phosphorylated Y198. Therefore, our LC-MS/MS data doesn’t support the notion that Panx1 is constitutively phosphorylated at Y198.

(3) The authors argue that the GFP tag of Panx1at the COOH terminus does not interfere with folding since the COOH modified (thrombin cleavage site) Panx1 folds properly, forming an amorphous glob in the cryo-EM structure. However, they do not show that the COOH-modified Panx1 folds properly. It may not, because functional data strongly suggest that the terminal cysteine dives deep into the pore. For example, the terminal cysteine, C426, can form a disulfide bond with an engineered cysteine at position F54 (Sandilos et al. 2012).

Our manuscript included results of using a non-GFP tagged PANX1 construct (Figure 2-figure supplement 1). We didn’t notice any difference for PANX1 phosphorylation between GFP-tagged and non-GFP-tagged PANX1. Therefore, the folding of the C-terminal tail of PANX1 doesn’t affect the conclusion of our study.

(4) The authors dismiss the additional arguments for tyrosine phosphorylation of Panx1 given by the various previous studies on Panx1 phosphorylation. These studies did not, as implied, solely rely on the commercial anti-phospho-Panx1 antibodies, but also presented a wealth of independent supporting data. Contrary to the authors' assertion, in the previous papers the pY198 and pY308 antibodies recognized two protein bands in the size range of glycosylated and partial glycosylated Panx1.

We didn’t dismiss additional arguments for the Src-dependent PANX1 regulation. In fact, in the discussion of our manuscript, we acknowledged the fact that Src may still be involved in PANX1 regulation, but probably through indirect mechanisms. In the two previous studies2,3, it’s unclear if the multimeric bands detected by pY198/pY308 antibodies correspond to glycosylated PANX1 or not, as the authors did not overlay the protein markers with their blots. In particular, the migration pattern of PANX1 changes across different western blot images from DeLalio et al2. It’s also worth noting that none of these “independent supporting data” in the two previous studies provided direct evidence that Src can phosphorylate pY198/pY308.

(5) A phosphorylation step triggering channel activity of Panx1 would be expected to occur exclusively on proteins embedded in the plasma membrane. The membrane-bound fraction is small in relation to the total protein, which is particularly true for exogenously expressed proteins. Thus, any phosphorylated protein may escape detection when total protein is analyzed. Furthermore, to be of functional consequence, only a small fraction of the channels present in the plasma membrane need to be in the open state. Consequently, only a fraction of the Panx1 protein in the plasma membrane may need to be phosphorylated. Even the high resolution of mass spectroscopy may not be sufficient to detect phosphorylated Panx1 in the absence of enrichment processes.

We agree with the reviewer that only plasma membrane-residing Panx1 phosphorylation is functionally relevant. Interestingly, however, previous studies actually analyzed total protein from cell lysate and concluded that PANX1 is phosphorylated at Y198 and Y308 (2,3). This has motivated our analysis, in which we found that the phosphorylation events cannot be detected when using whole cell lysate. Therefore, we have also conducted an electrophysiology experiment by comparing conditions with/without active Src kinase (Figure 7). Our result indicates that PANX1 current is not affected by the presence of Src. This result suggests that even if there might be minor Src kinase phosphorylation beyond detection limit of western blot or mass spectrometry, they may not be functionally significant as well.

(6) In the electrophysiology experiments described in Figure 7, there is no evidence that the GFP-tagged Panx1 is in the plasma membrane. Instead, the image in Figure 7a shows prominent fluorescence in the cytoplasm. In addition, there is no evidence that the CBX-sensitive currents in 7b are mediated by Panx1-GFP and are not endogenous Panx1. Previous literature suggests that the hPanx1 protein needs to be cleaved (Chiu et al. 2014) or mutated at the amino terminus (Michalski et al 2018) to see voltage-activated currents, so it is not clear that the currents represent hPANX1 voltage-activated currents.

Our previous analysis has already shown that endogenous current of non-transfected cells is not sensitive to CBX (4). Therefore, the CBX-sensitive current in cells overexpressed PANX1 is from PANX1-GFP. It should be noted that when protein is overexpressed, it tends to accumulate at different intracellular membranes during protein synthesis/maturation. However, this doesn’t affect a portion of the protein to be trafficked to the plasma membrane. In the paper from Michalski et al 2018, it was shown that WT human/mouse PANX1 displayed voltage-dependent activation (5). Although the current is relatively small, it is clearly distinguishable from non-transfected HEK and CHO cells. This voltage-dependent activation is also sensitive to CBX, consistent with our measurement (Figure 7) (4). When GS is introduced at the N-terminus, the voltage-dependent activation of human/mouse PANX1 is significantly boosted, likely due to the altered NTH conformation resulting from the N-terminal extension.

**Recommendations for the authors:**

**Reviewer #3 (Recommendations For The Authors):**
Literature quotes are still problematic. Why are secondary papers quoted instead of the original work? At least quote reviews by authors who published the original findings.

We appreciate the reviewer pointing this out. We have carefully checked our references and made sure that the original literature is cited.

Why does wtPanx1 run close to the 37 kD marker (Figure 2 supplement 1) instead of close to 50 kD as shown in the previous papers using the pY198 and pY308 antibodies?

It is a common observation that membrane proteins migration in SDS-PAGE gel doesn’t correlate with their formula molecular weight, also known as “gel shifting” (6–8). The molecular mechanism of this phenomenon remains complex. Therefore, simply relying on protein molecular standard could not unambiguously identify PANX1 protein band. This is an issue for identifying PANX1 band, especially in light of the fact that some antibodies may not be very specific (see Figure 6B). In our experiment, we have correlated the in-gel fluorescence and western blot signal which allowed us to determine the protein band corresponding to PANX1. It is worth noting that in Figure S3 of DeLalio 2019, the PANX1 is detected at 37 kDa (2). However, in many other panels of the paper, PANX1 is detected at close to 50 kDa (for example, Figure S2B).

Figure 6, supplement 1: why are there oligomers observed in the absence of crosslinking? Why is there no shift in the size of the "oligomers" in response to glycosidase F?

It is common to observe multimeric membrane proteins, including PANX1, forming oligomeric bands in SDS-PAGE gels, likely because they are not fully denatured or disassembled. PANX1 also contains several free cysteines, which may non-specifically crosslink subunits. There is actually a small shift for the 75 kDa band (dimer) in Figure 6, supplement 1. For higher molecular weight bands, this small shift may not be apparent due to the limited resolution of the gel.

A positive control for the antibodies used is missing. The authors argue that such controls are not available, since these commercial antibodies are "proprietary".

We did provide several positive controls for the antibodies in our study. We showed that the anti-PANX1 and anti-Src antibodies unambiguously recognized PANX1 and Src, respectively (Figure 3A), and that a pan-specific phosphotyrosine antibody (P-Tyr-100) unambiguously recognized phosphorylated Src (Figure 3A)—as expected—but did not recognize PANX1. In addition, we demonstrated that the two antibodies in question (anti-PANX1-pY198 and anti-PANX1-pY308) did produce signals in our western blot analysis, but we provided compelling evidence that the bands produced by these antibodies do not correspond to PANX1 (Figure 2B).

Unfortunately, the epitopes that Millipore Sigma used to generate anti-PANX1-pY198 and anti-PANX1-pY308 are not available. The description of the immunogen from Millipore Sigma website states that “A linear peptide corresponding to 12 amino acids surrounding phospho-Tyr198 of murine Pannexin-1” and “A linear peptide corresponding to 13 amino acids surrounding phosphotyrosine 308 of rat pannexin-1”. However, these immunogen peptides are not available for us to purchase.

References

(1) Nouri-Nejad, D. et al. Pannexin 1 mutation found in melanoma tumor reduces phosphorylation, glycosylation, and trafficking of the channel-forming protein. Mol Biol Cell 32, (2021).

(2) DeLalio, L. J. et al. Constitutive SRC-mediated phosphorylation of pannexin 1 at tyrosine 198 occurs at the plasma membrane. Journal of Biological Chemistry 294, (2019).

(3) Weilinger, N. L. et al. Metabotropic NMDA receptor signaling couples Src family kinases to pannexin-1 during excitotoxicity. Nat Neurosci 19, (2016).

(4) Ruan, Z., Orozco, I. J., Du, J. & Lü, W. Structures of human pannexin 1 reveal ion pathways and mechanism of gating. Nature 584, (2020).

(5) Michalski, K., Henze, E., Nguyen, P., Lynch, P. & Kawate, T. The weak voltage dependence of pannexin 1 channels can be tuned by N-terminal modifications. Journal of General Physiology 150, (2018).

(6) Rath, A., Cunningham, F. & Deber, C. M. Acrylamide concentration determines the direction and magnitude of helical membrane protein gel shifts. Proc Natl Acad Sci U S A 110, (2013).

(7) Rath, A. & Deber, C. M. Correction factors for membrane protein molecular weight readouts on sodium dodecyl sulfate-polyacrylamide gel electrophoresis. Anal Biochem 434, (2013).

(8) Rath, A., Glibowicka, M., Nadeau, V. G., Chen, G. & Deber, C. M. Detergent binding explains anomalous SDS-PAGE migration of membrane proteins. Proc Natl Acad Sci U S A 106, (2009).